# Quantification of Fluid Volume and Distribution in the Paediatric Colon via Magnetic Resonance Imaging

**DOI:** 10.3390/pharmaceutics13101729

**Published:** 2021-10-19

**Authors:** Jan Goelen, Benoni Alexander, Haren Eranga Wijesinghe, Emily Evans, Gopal Pawar, Richard D. Horniblow, Hannah K. Batchelor

**Affiliations:** 1School of Pharmacy, Institute of Clinical Science, University of Birmingham, Edgbaston, Birmingham B15 2TT, UK; JXG918@student.bham.ac.uk (J.G.); G.Pawar@bham.ac.uk (G.P.); 2School of Biomedical Science, Institute of Clinical Science, University of Birmingham, Edgbaston, Birmingham B15 2TT, UK; benoni.alexander@gmail.com (B.A.); R.Horniblow@bham.ac.uk (R.D.H.); 3Department of Radiology, University Hospital Coventry & Warwickshire, Clifford Bridge Road, Coventry CV2 2DX, UK; haren_wijesinghe@hotmail.com (H.E.W.); emilylivesey58@gmail.com (E.E.); 4Strathclyde Institute of Pharmacy and Biomedical Sciences, University of Strathclyde, Glasgow G4 0RE, UK

**Keywords:** intestinal fluid, MRI, paediatrics, biorelevant dissolution, large bowel, colon

## Abstract

Previous studies have used magnetic resonance imaging (MRI) to quantify the fluid in the stomach and small intestine of children, and the stomach, small intestine and colon of adults. This is the first study to quantify fluid volumes and distribution using MRI in the paediatric colon. MRI datasets from 28 fasted (aged 0–15 years) and 18 fluid-fed (aged 10–16 years) paediatric participants were acquired during routine clinical care. A series of 2D- and 3D-based software protocols were used to measure colonic fluid volume and localisation. The paediatric colon contained a mean volume of 22.5 mL ± 41.3 mL fluid, (range 0–167.5 mL, median volume 0.80 mL) in 15.5 ± 17.5 discreet fluid pockets (median 12). The proportion of the fluid pockets larger than 1 mL was 9.6%, which contributed to 94.5% of the total fluid volume observed. No correlation was detected between all-ages and colonic fluid volume, nor was a difference in colonic fluid volumes observed based on sex, fed state or age group based on ICH-classifications. This study quantified fluid volumes within the paediatric colon, and these data will aid and accelerate the development of biorelevant tools to progress paediatric drug development for colon-targeting formulations.

## 1. Introduction

Oral drugs are the most common form of drug administration due to both patient convenience and the favourable cost:benefit ratio [1,2]. Despite the clear advantages of oral formulations, such preparations rely on drug liberation and dissolution for absorption. The free water available at the site of dissolution is a critical parameter for these processes as it also affects local drug concentration and thus permeation [3,4,5]. As such, a comprehensive understanding of the amount and distribution of fluid throughout the gastrointestinal tract (GIT) is required to ensure appropriate and adequate absorption of oral medicines [6]. This is important for colon-targeted formulations, which have gained increased interest from the pharmaceutical industry [7,8,9], for local action or systemic absorption [10,11,12]. Increasing the absorption window of a drug by targeting the colon offers advantages in terms of frequency of dosing, where controlled or extended release (XR) formulations can be used to enable once daily dosing (improving patient convenience [8], maintaining therapeutic concentrations [13] and reducing the risk of administration errors [14]). The low proteolytic activity and the potential of intact peptide absorption [15] (as demonstrated for insulin [16] or linaclotide [17]) enables the large intestine (with emphasis on the proximal colon) as an appropriate absorption site [5,18].

Colon-targeting formulations are often designed to exploit local intestinal environment characteristics for drug release, such as pH or the presence of bacterial-derived metabolising enzymes [10,12]. Although some XR-formulations are designed to deliver drugs independent of local environmental factors (such as time-dependent release, such as mesalazine preparations for ulcerative colitis treatment [19]), the performance of these formulations needs to be evaluated in biorelevant conditions that include the colonic macroenvironment in terms of fluid volume and composition [14,20,21,22]. In addition, the influence of colonic fluid volumes on incomplete dissolution and absorption of poorly soluble drugs in the upper GIT and subsequent accumulation of solid drug particulates in the colon [23,24] (such as NSAIDs for treatment of colorectal cancer) is not fully understood.

Data on intestinal fluid (amount, distribution and composition) are necessary for in vitro and in silico models in drug development [3,5,8,11,14,25,26]. Despite recent efforts, there is no consensus on the volume of water or its distribution throughout the colon of adults [27], which translates in poor estimations for the standardised volume employed for colonic dissolution testing (Table 1). Currently, in vitro and in silico models of the colon use a range of volumes for dissolution assays, ranging from 1 mL to 200 mL [28,29,30].

As XR-formulations are associated with high inter-individual variability in vivo [33,34], it is recognised that prediction of performance of these products is challenging. The uncertainty is even more pronounced for paediatric XR-formulations, as the variability is increased due to the lack of physiologically relevant input data for in vitro and in silico models [2,35].

There is a recognised knowledge gap about the amount and distribution of free fluid in the GIT of the paediatric population [36,37,38], resulting in a need to generate physiological data to underpin the development of age appropriate physiologically based pharmacokinetic (PBPK) models. This would improve prediction of drug performance in children, as paediatric clinical trials often face ethical constraints [1,2]. No studies to date have quantified the amount and distribution of fluid in the paediatric colon [36,37,38,39], thus predictions of performance are based on extrapolation of adult data. Direct extrapolation of the adult values to the paediatric population is less appropriate, due to the reported differences in the paediatric anatomy and physiology of the GIT [1,40,41].

The free fluid in the GIT can be quantified by magnetic resonance imaging (MRI), a non-invasive tool that permits undisturbed visualisation of the GIT [7]. In contrast with scintigraphy or computed tomography (CT), no ionising radiation dose is needed for imaging [27] and its effectiveness in producing qualitative images enabling fluid quantification has been demonstrated [3,27,31,32,42,43].

Therefore, the aim of this study was to use MRI to locate and quantify the fluid volumes and number of pockets in the colon of a paediatric population.

## 2. Materials and Methods

### 2.1. Study Design and Participants

Our research group has access to an MRI databank showing the abdomen of 49 paediatric participants, which was used previously to quantify the volume of free fluid in the stomach and small intestine. This observational, retrospective study used the same MRI datasets to quantify the volume of free fluid in the colon (Ethical approval: REC reference: 18/EM/0251 (IRAS 237159 MRI: Fluid volumes and localisation in paediatric GI tract)). Following exclusion criteria were applied to ensure the cohort was as healthy as possible: patients with acute abdomen (appendicitis or perforated viscus), malignant bowel disease, surgery (bowel section, excluding appendicectomy), bowel wall thickening/stricture/fistula/abscess. The datasets originated from two sites: Birmingham Children’s Hospital (BCH) and University Hospitals Coventry and Warwickshire NHS Trust (UHCW). All participants were fasted overnight and in addition, the UHCW site required the children to ingest 500 mL of Oral Klean Prep (a macrogol-based osmotic laxative) 60 min before MRI acquisition. This enabled the study to include a fasted and a fluid-fed population. The MRI acquisition parameters are listed in Table 2.

### 2.2. Data Processing

Only datasets with good resolution and correct T2-weighted MRI sequence were included in the study; this was a total of 46 MRI acquisitions. A 2D- and 3D-model were used to quantify the fluid within the MRI datasets.

#### 2.2.1. D-Protocol

Two software packages were used for the 2D-fluid volume determination (Figure 1):Horos [44] to identify and highlight the fluid pockets in the MRI dataset;ImageJ to calculate the area of the marked fluid regions.

Horos is an open-source code software program distributed free of charge under the LGPL license at Horosproject.org and sponsored by Nimble Co LLC d/b/a Purview in Annapolis, MD, USA. Fluid identification within the MRI slices was based on the intensity of cerebrospinal fluid (CSF) which is routine in interpretation of these images [3,4,31,42,45]. The average voxel intensity within the CSF was used as threshold value for free fluid. The plug-in Global Thresholding Tool [46] then identified and highlighted all zones in the entire MRI dataset with a voxel intensity equal or higher than the threshold value (thus representing free fluid) in red (Figure 1B). These marked images were transferred to ImageJ [47], where the two dimensional area of the highlighted zones were calculated (Figure 1C,D). The volume of each zone was calculated by multiplying the highlighted area by the sum of the slice thickness and the interstitial slice gap. The volume, number and location of the fluid pockets were recorded for every participant. The location (ascending, transverse or descending colon) of fluid pockets were manually determined [48]. Paediatric radiologists assisted in defining the pocket’s location. The first image slice (when migrating dorsal to ventral) showing the hepatic flexure is where the transverse colon was determined to start (on both the ascending-transversal and transversal-descending junctions). This enabled the exclusion of areas that were motion artefacts [49]. A sub-set of 20% were analysed by a second operator.

#### 2.2.2. Three-Dimensional Protocol

Two software packages were used to calculate the number of fluid pockets in addition to quantifying the free fluid volume via a 3D-model (Figure 2):Horos to identify and highlight the fluid pockets in the MRI dataset;Blender to remove artefacts and isolate and compute the fluid pockets.

In order to measure individual fluid pockets within the colon, the MRI dataset after the CSF-based thresholding was converted into a 3D-model. Initially, the 3D-volume rendering feature within Horos was used (Figure 2A), which takes the slice thickness and interstitial slice gaps into account. Subsequently, within the colon, the highlighted free fluid areas were approximately excised using the scissor tool, resulting in a 3D-model showing only the fluid pockets (Figure 2B). This model was then exported as a stereolithography (STL) file (Figure 2C). The software programme Blender was used to manipulate the STL file. Blender is a free and open-source 3D-creation suite, released under the GNU General Public License. Blender was used to refine the excision of the colon and remove any non-colon remaining artefacts within the 3D-model, as well as to calculate the volume of each individual pocket. After fixing the scale within Blender to the scale of the MRI image, the fluid pockets volumes were measured by using the plug-in “Mesh—3D print toolbox” in Blender (Figure 2D). The volume was measured for each individual fluid pocket via the volume feature. The sum of all individual fluid pockets gave the number of pockets for the entire dataset, as well as a second (alternative) measurement of the fluid volume within the paediatric dataset.

### 2.3. Statistical Analysis

SPSS [50] was used for statistical analysis. A statistical test was deemed significant if the *p*-value (*p*) was smaller than 0.05. To investigate correlations, Pearson’s correlation tests were used. To compare between two groups (such as investigating feed state), independent sample *t*-tests were used. Where multiple groups were compared, a one-way ANOVA with Bonferroni correction was used. Where multiple measurements on the same participants were investigated (such as a difference in volume in the three colon segments), an ANOVA of repeated measures with a Greenhouse–Geisser correction and post hoc Bonferroni correction was used.

## 3. Results

### 3.1. Participant Demographics

The participants MRI datasets were stratified into age groups according to the International Council for Harmonisation of Technical Requirements for Pharmaceuticals for Human Use (ICH) classifications [51,52] (Table 3) with the cohort younger than 2 years old named as infants collectively. Detailed metadata (specific age, weight and sex) for six participants were missing, although their ICH age group was known. Subsequently, these datapoints were excluded from analyses that included sex or specific age. Furthermore, four more datasets were excluded from the 3D-based determination of fluid volume and number of pockets, as they were incompatible with the Blender protocol. Data on threshold values, fluid volumes via both protocols and number of pockets in the total and segments of the colon per participant is available in Appendix A.

### 3.2. Colonic Fluid Volume and Number of Pockets

The paediatric colon contained an average of 22.5 (±41.3 standard deviation) mL of fluid in 15.5 (±17.5 standard deviation) discreet fluid pockets; the data were not normally distributed, and the median volume was 0.80 mL with a median number of 12 pockets (Table 4).

The majority of the fluid pockets were small (i.e., volume smaller than 1 mL); a total of 90.4% of the pockets (557/616) were smaller than 1 mL, which accounted for 5.5% of the total fluid volume (51.5 mL/933.2 mL) in the database. The other 9.6% of the pockets (59/616) were larger than 1 mL and contributed 94.5% to the total fluid volume (881.7 mL/933.2 mL) in the database. The paediatric colon contained on average 3.6 ± 2.9 pockets larger than 1 mL (Table 5).

There was a large variation observed in the datasets, both in fluid volume and number and size of pockets. No colonic fluid was found in nine participants, whereas eight datapoints with a total fluid volume higher than 60 mL were classified as statistical outliers, however these were not excluded from the subsequent analysis (shown as open symbols in (Figure 3A). The nine participants with no visible fluid had no common demographic factor (age group, sex, fed state), neither did the eight participants with a colon fluid volume higher than 60 mL.

There was a trend for decreased fluid volume in the segments towards the distal colon (Figure 3B). The fluid volumes of the colonic ascending, transverse and descending segments were positively correlated to each other (Pearson’s coefficient >0.58, *p* < 0.001 in all cases). ANOVA of repeated measures with a Greenhouse–Geisser correction and post hoc Bonferroni correction showed that the ascending colon contained the most fluid (*p* < 0.001). No statistically significant difference was detected between the fluid volumes in the two other segments. However, in four MRI datasets (with the total volume ranging from 0.004 mL to 0.4 mL) either the transverse or descending colon contained all the fluid (and thus none was present in the ascending colon).

The ascending colon contained the most fluid pockets compared with the other two segments (*p* < 0.001); no statistically significant difference was found in the number of fluid pockets between the transverse and descending colon. Only the ascending colon contained fluid pockets larger than 1 mL, apart from one dataset where a pocket of 249.6 mL was observed that spanned the entire colon. A graphical representation of the distribution of the individual pockets in the entire colon with a volume of up to 25 mL is shown in Figure 4.

No correlation was detected in the entire cohort between age and the fluid volumes or number of pockets in the total colon or segments (*p* > 0.17 in all cases, Figure 5A). However, in the preschool population (2–5 years old, *n* = 11), age was positively correlated (Pearson’s coefficient >0.65) to total and ascending colonic fluid volume (for both *p* < 0.03, Figure 5B), but no correlation was found between age and the number of fluid pockets.

No significant differences in fluid volumes or the number of pockets were detected based on fed state, sex or age group.

### 3.3. Robustness of Protocol

There was good similarity in threshold values and total fluid volumes between the two operators (Pearson’s correlation >0.97 and *p* < 0.001 in both comparisons, Appendix A), demonstrating the absence of operator-bias. Furthermore, the fluid volumes obtained via the ImageJ protocol were in good correlation to the 3D-based protocol (Pearson’s correlation 0.94, *p* < 0.001, Appendix A). There were no correlations between threshold value and any of the fluid volumes or the number of pockets, demonstrating robustness of the followed protocol (Appendix A).

## 4. Discussion

This study is the first to report fluid volumes in the paediatric colon using MRI quantification. The median fluid volume found in the total colon was 0.80 mL, the mean 22.48 mL ± 41.30 mL, with a range of 0 mL to 167.5 mL (Table 4). A comparison of these data to other paediatric colonic volumes could also not be made due to a lack of available data [36].

The median paediatric colonic fluid value (0.80 mL) is considerably lower than that observed in adults (2 mL [32] or 8 mL [31]), indicating the colonic fluid volumes are different in children. However, the mean results from this paediatric study are comparable to values reported for the adult colon (Figure 6), which was surprising as the physiology of the colon changes with age.

The median total colonic volume (TCV) of healthy children aged 14–18 years is reported to be 227 mL (interquartile range 180 mL–263 mL) [53], whereas the mean healthy adult TCV is 760 mL (with a 95% confidence interval of 662 mL–858 mL) [27,54]. Similarly, the length of the colon increases with age, from 52 cm length in children younger than 2 years of age to 150 cm in adults [55]. Therefore, a smaller colonic free fluid volume in the paediatric colon compared with an adult cohort is expected. However, not all colonic parameters change with maturation, e.g., the pH of the paediatric colon children aged 8–14 years is reported to be comparable to adult values [56,57]. Similarly, no significant difference was detected in the fluid volumes between the four different paediatric age groups, (consistent with the existing data on small intestinal fluid volumes) in children [4].

As the generic ICH-classification is based on days/months/years after birth, stratification on a parameter that considers the physicochemical properties of GIT anatomy and contents might have been more appropriate, such as height, body surface area (BSA) or body mass index (BMI) as these are typically used for dose adjustments. However, such data were not collected from the participants. A limitation in this study is that data were only available from 49 patients, extracted from two tertiary centres with differing MRI acquisition protocols. Paediatric MRI small bowel is a scarce resource offered mainly in tertiary centres (UHCW and BCH). It is a specialist modality only performed on selected cases, following discussion at a small bowel multidisciplinary meeting and requires interpretation by specialists. Therefore, the number of paediatric small bowel available for scientific study is small when compared with their adult counterparts. Together, these factors led to a small sample size and even smaller sub-sample size, when comparisons were made based on fed-status, sex or age; in addition to this, the numbers of participants in these sub-groups were not well balanced that may introduce bias into the statistical analysis and interpretation from these groups. Although the sample size is statistically small, it is sizeable considering population type and complexity of the modality; as such, these data should be viewed as preliminary where a larger, well defined cohort is warranted to provide robust data that will lead to more conclusive claims.

As the influence of age was investigated by comparing four paediatric age groups, the sample size per age group was similar to sample sizes used in the previous studies on healthy adult volunteers. However, the demographics of the participants in the adult studies were closely controlled; for example, those in Murray et al. [3] (*n* = 12) were all 20–22 years old and healthy, to minimise variability in the dataset. The paediatric participants have potential underlying morbidities, resulting in variability which may have introduced bias into the data. The large variability in colon fluid volumes was also observed in studies on healthy adults via MRI quantification [32,42,58,59]. The variability in studies undertaken using a small group of healthy volunteers in a clinical setting are less likely to capture the real-world variability, compared with studies in a heterogenic paediatric population with potential comorbidities [38,60].

There were on average 3.6 ± 2.9 fluid pockets larger than 1 mL, mainly in the ascending colon. Similarly to adults, the majority of the fluid pockets were very small [3] (Figure 4 and Figure 7). In this study, it was found that only 9.6% of the fluid pockets were larger than 1 mL, yet they contributed for 94.5% to the overall volume quantified. The software protocols used in this study did not use any limits on the size of fluid pockets to be quantified, therefore even fluid pockets of 0.01 mL were included in the data. As 90.4% of the fluid pockets were smaller than 1 mL and contributed to only 5.5% of the fluid observed in the study (55.5 mL), their physiological relevance in colonic absorption might be questioned.

No correlation was observed between colonic fluid volumes or the number of pockets, and age or weight. This implies that age or weight does not correlate to colonic fluid volumes, which is in contrast with other GIT parameters such as gastric pH [1] or the need for fluid intake [51] where the most significant changes are observed in neonates and infants (<2 years) [61]. Current dosage guidelines use allometric scaling based on age, weight or BSA to extrapolate paediatric doses from adult data [37,51,62,63,64,65]. Further investigation is required to investigate the appropriateness of allometric scaling for colon-specific drug delivery systems colon (CDDS) [37], especially for poorly soluble drugs that act within the colon. Only in the preschool category (children aged 2–5 years, *n* = 12), was age significantly positively correlated to fluid in the total and the ascending colon (Figure 5B). This implies that children in this category will have more fluid in their ascending (and thus total) colon as they are closer to 5 years of age; this finding could be due to the small sample size and the large variability seen within these 12 participants.

Fed status had no impact on fluid volumes (data not shown), which could be a result of several factors. Firstly, it was unlikely that Oral Klean Prep had an impact on the colon fluid volumes in the timeframe of this study, although an effect could not be excluded a priori, as literature is conflicting about when an ingested solution reaches the colon since this is subjected to inter- and intra-variability [27]. In addition, Oral Klean Prep is an osmotic laxative, which draws water into the GIT. Therefore, it is not fully representative to a fluid-fed child. Placidi et al. [66] reported a significant increase in ascending colon fluid volumes 45 min post-ingestion [66] of 5% mannitol (another osmotic laxative) in 350 mL of water, so a possible effect could not be excluded in advance. The healthy paediatric population is believed to have a gastric emptying rate [67] and small intestinal transit time (SITT) comparable to adults (SITT 3.49 h ± 1.02 h (mean ± SD)) based on meta-analysis [38,65,68], although data suggest children younger than 2 years of age have a slower (longer) SITT [38,51]. In addition, this meta-SITT was not affected by fed state [68]. The meta-gastric emptying rate and meta-SITT values support the absence of an impact of Oral Klean Prep on the colon volumes as observed in this study. The increase of the free fluid in the colon of fasted adults Pritchard et al. [32] observed 60 min post-ingestion of 500 mL Moviprep or Murray et al. [3] noticed 30 min post-ingestion of a 240 mL glass water is likely to be due to the gastrocolonic reflex, as hypothesised by Lemmens et al. [69]. Secondly, there was no verification of fed state prior to analysis as this study had a real-world setting. The consumption of the full 500 mL Oral Klean Prep was not monitored. In addition, it was not strictly monitored whether the children in the fasted population indeed fasted overnight, although this was part of their clinical instructions. Tighter monitoring on the clinical protocols prior to MRI with specific reference to fasting and ingestion of the solution would improve the interpretation of this MRI data.

The ascending colon typically contained the highest portion of fluid (Figure 3). Correlations demonstrated that when fluid is present in the ascending colon, this can act as a predictor for the total colonic volume. The same trend was observed in adults [3,27]. Consequently, as chyme from the small intestine enters the colon, the desiccating function of the colon to transform chyme to drier stools explains this trend [36], where less water is identified in more distal regions. It should be noted that only the free-flowing water is being quantified, based on the fluidity of the CSF [42].

There is substantial variation in the choice of software used to interrogate MRI datasets in order to calculate fluid volumes and pockets [3,4,31,32,42,45,58]. Previous studies have already expressed the need for standardisation of methods [26], as the CSF-based threshold for free fluid is dataset dependent. However, no influence of CSF-threshold values to the obtained data in this study was detected, consistent with literature [3,4,31,42]. In addition, cross-analysis of the extracted values between the 2D- and 3D-based protocol and between multiple operators shows that both protocols produce similar results regardless of operator or used modelling approach (Appendix A). Therefore, the use of CSF-based thresholds on the MRI datasets are robust and provide consistent results for colonic fluid analysis.

This pioneering study quantified the free fluid in the paediatric colon, which is of great interest to the pharmaceutical industry [27] for paediatric drug development as clinical studies in the paediatric population often face ethical restrictions [30,36]. Although most data currently available for in silico models are derived from healthy Caucasian adults, accurate physiological data derived from the intended patient population are more appropriate for biorelevant modelling [35,70], e.g., to account for an altered GIT physiology due to disease or age [38]. For the first time, in vitro and in silico models can be developed tailored to the paediatric cohort [36,38] that are informed by real-world data on the volume of fluid (mean, median, extreme values and variability) within the paediatric colon.

## 5. Conclusions

This study successfully quantified the fluid volumes within the paediatric colon. Two methods were employed to quantify fluid in MRI datasets and their robustness was demonstrated via cross-analysis between operators and methods.

The small overall sample size and even smaller sub-population sizes meant that these data are preliminary, and a fuller cohort study is required to verify the findings presented here. The median fluid volume for the total paediatric colon (0.80 mL) is comparable to literature data for the adult colon, although the mean paediatric volume (22.48 ± 41.30 mL) is nearly double than the adult value standardly used, i.e., 13 mL. The high variability was also observed in adults. No overall correlation was detected between colonic fluid volumes and age, similar to results in the paediatric SI. Fed status, sex or age across the whole population did not influence the colon fluid volumes. The fact that a significant correlation was observed in the 2–5 years group (who were all fasted) between age and fluid volumes in the entire and ascending colon warrants further investigation. No such correlation was observed for other age groups. Furthermore, the ascending colon contained the most fluid compared with the transverse and descending colon. This study demonstrates the feasibility of obtaining real-world data from MRI to inform physiologically based models, which minimises the burden to special populations.

The novel output from this study will improve the physiological understanding of the paediatric colon, aid biopredictive in silico simulations and establishing novel, more accurate in vitro assays and thus support paediatric drug development that targets the colon, resulting in more age-appropriate medicines for the paediatric population [71,72].

## Figures and Tables

**Figure 1 pharmaceutics-13-01729-f001:**
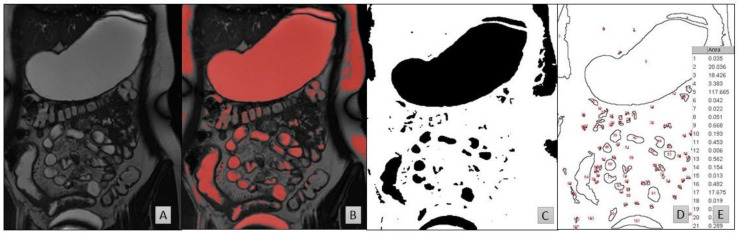
MRI slices taken from a 16-year-old fed female. (**A**) The original MRI slice. (**B**) The same slice after thresholding in Horos based on the CSF. (**C**) The same slice after filtering the red pixels on a black-on-white print in ImageJ. (**D**) The results of particle analysis show the outlines of the regions of interest and (**E**) an example extract of the data generated following calculation of the respective areas from (**D**).

**Figure 2 pharmaceutics-13-01729-f002:**
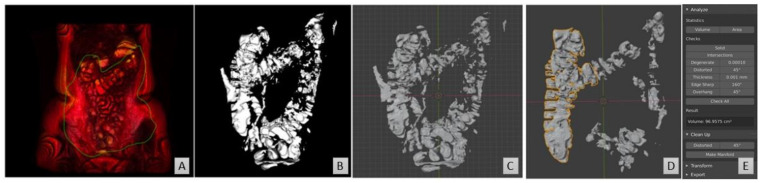
Representative images from the 3D protocol, MRI dataset taken from a 15-year-old female. (**A**) The 3D rendering in Horos builds a 3D model of the entire dataset. (**B**) The colon was excised from the 3D model in Horos. (**C**) The model converted into an STL file and opened in Blender. (**D**) Non-colon artefacts are removed. (**E**) The volume of the individual pockets is calculated in Blender.

**Figure 3 pharmaceutics-13-01729-f003:**
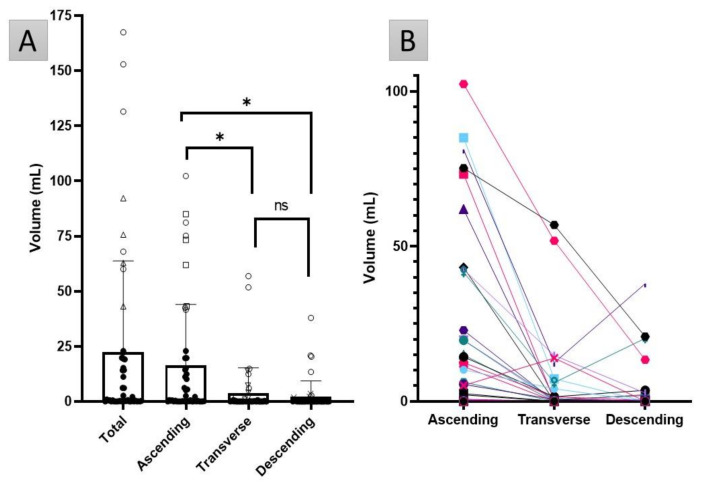
(**A**) The fluid volume in the total colon and the ascending, transverse and descending segments. The open symbols indicate the individual datapoints that are statistical outliers: circles are outliers in every section, the other symbols indicate outliers only in that particular segment (triangles in the total colon: squares for the ascending colon, triangles for the transverse colon and crosses for the descending colon). The bar chart shows mean values, with standard deviation as the error bar. The * represents a significant difference (*p* < 0.05); ns = not significant. (**B**) Colon fluid volume in each segment of the colon, linked per participant.

**Figure 4 pharmaceutics-13-01729-f004:**
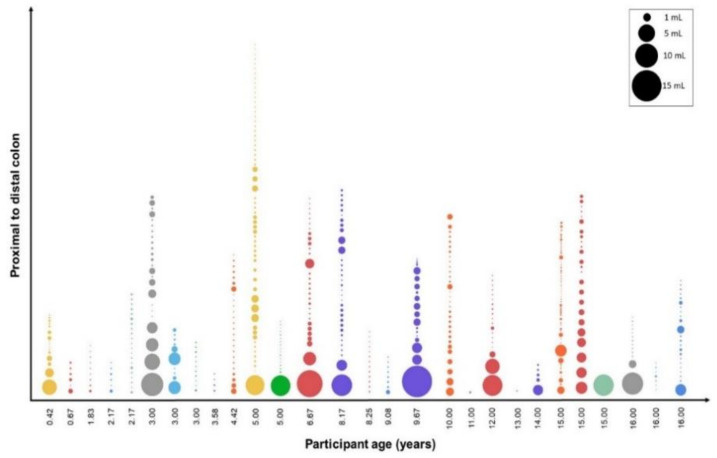
Representation of fluid pockets in the paediatric colon for each participant arranged in order of increasing age. Each bubble depicts an individual fluid pocket, the size illustrates the relative volume. The nine pockets larger than 25 mL, found in nine individual datasets, are not shown, as they distort the scale.

**Figure 5 pharmaceutics-13-01729-f005:**
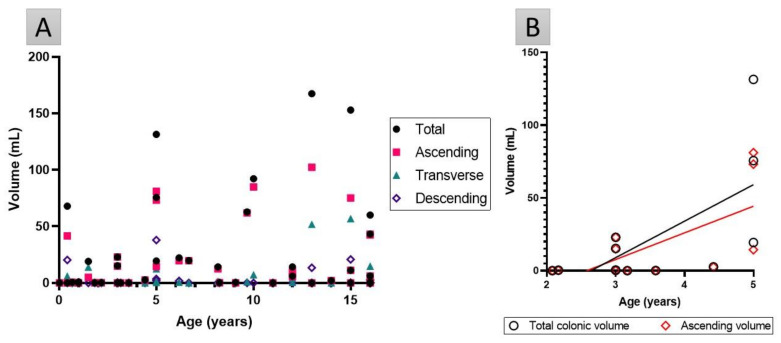
(**A**) Fluid volume in the total and segments of the colon for every participant as a function of age. No correlations were found, (*p* > 0.17 in all cases). (**B**) Fluid volume in the total and ascending colon in the preschool population are significantly correlated to age (Pearson’s coefficient >0.65, *p* < 0.03).

**Figure 6 pharmaceutics-13-01729-f006:**
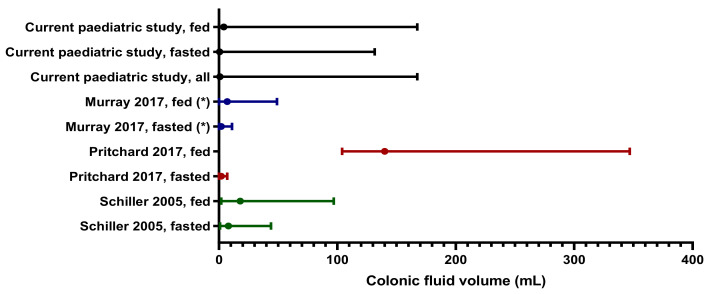
Comparison of the median (closed dot) with associated range reported for colon volumes in adult studies with healthy volunteers. (*): Note that for Murray et al., 2017, the mean instead of median is reported.

**Figure 7 pharmaceutics-13-01729-f007:**
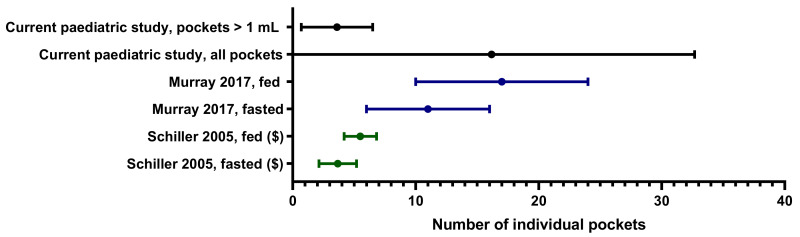
Comparison of the mean and SD reported for the number of fluid pockets in the colon in adult studies with healthy volunteers. ($): Note that for Schiller et al., 2005, the 25% and 75% percentiles are reported instead of the SD.

**Table 1 pharmaceutics-13-01729-t001:** Reported fluid volumes in the colon of healthy adults when measured using MRI. N/A means that the data were not reported.

Study	Feed Status (Intake of Food/Fluid)	Time of Ingestion before MRI Acquisition	Number of Participants	Median (Min-Max) (mL)	Mean (±SD) (mL)
Schiller 2005 [31]	Fasted	-	12	8 (1–44)	13 ± 12
	Fed (standardised meal)	1 h		18 (2–97)	11 ± 26
Pritchard 2017 [32]	Fasted	-	11	2 (0–7)	-
	Fed (500 mL Moviprep)	1 h		140 (104–347)	-
Murray 2017 [3]	Fasted	-	12	N/A (0–11)	2 ± 1
	Fed (240 mL water)	30 min		N/A (0–49)	7 ± 4

**Table 2 pharmaceutics-13-01729-t002:** MRI scanner and acquisition parameters.

Site	UHCW	UHCW	BCH	BCH
Participants	Fluid-fed	Fluid-fed	Fasted	Fasted
1.5 T MR Imaging Unit: series and manufacturer	Optima MR450w, GE Healthcare, Chicago, IL, USA	Aera, Siemens Healthcare, Erlangen, Germany	Siemens MAGNETOM Avanto 1.5 T MRI System, Siemens Healthcare, Erlangen, Germany	Aera, Siemens Healthcare, Erlangen, Germany
MRI coil	48-channel body coil	body coil	16-element parallel imaging receiver coil	16-element parallel imaging receiver coil
MRI protocol	Coronal balanced steady-state gradient echo sequence (fast-imaging employing steady-state acquisition, FIESTA)	Coronal balanced steady-state gradient echo sequence (true FISP)	Coronal T2 SPACE sequence	Coronal T2 SPACE sequence
Median slice thickness (range)	4.0 mm (2.998 mm–6 mm)	6.0 mm (2.998 mm–6 mm)	0.9 mm (0.09 mm–0.55 mm)	0.9 mm (0.09 mm–0.55 mm)
Echo train length	1	1	1	1
Median intersection gap	5.0 mm	3.0 mm	None	None
Matrix	0.35 × 0.35 mm	1.0 × 1.0 mm	0.8 × 0.8 mm	0.8 × 0.8 mm
Field of view	420 cm^2^	420 cm^2^	250 cm^2^	400 cm^2^
TR/TE	5.7/1.9 ms	652.8/2.1 ms	1700/98 ms	2000/241 ms

**Table 3 pharmaceutics-13-01729-t003:** Demographics of the participants included in this study [4].

Age Range	Number of Datasets Available	
	Fasted Children	Fluid-Fed Children
<2 years (neonate/infant/toddler)	9	0
2–5 years (pre-school children)	12	0
6–11 years (school-age children)	6	2
12–16 years (adolescents)	1	16

**Table 4 pharmaceutics-13-01729-t004:** Fluid volumes and number of pockets for the total, ascending, transverse and descending colon.

Colon Segment	Total	Ascending	Transverse	Descending
Mean ± SD (mL)	22.48 ± 41.30	16.44 ± 27.62	3.78 ± 11.49	2.27 ± 7.09
Median (mL)	0.80	0.63	0.004	0.003
Interquartile range (mL)	19.69	18.52	0.65	0.12
Range (min-max) (mL)	0–167.47	0–102.30	0–56.87	0–37.94
Mean ± SD number of fluid pockets	15.5 ± 17.5	14.5 ± 16.4	1.0 ± 2.5	0.05 ± 0.2
Median number of fluid pockets	12	10	0	0

**Table 5 pharmaceutics-13-01729-t005:** Mean and median number and volumes of pockets per participant.

	All Pockets		Pockets Bigger than 1 mL	
	Number per Participant	Volume (mL)	Number per Participant	Volume (mL)
Mean ± SD	15.5 ± 17.5	1.52 ± 12.09	3.6 ± 2.9	14.60 ± 36.29
Median	12	0.04	3	3.31

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
