# Peer review of "Quantification of Fluid Volume and Distribution in the Paediatric Colon via Magnetic Resonance Imaging"

_pharmaceutics, 2021, doi:10.3390/pharmaceutics13101729_

Round 1

Reviewer 1 Report

This study focuses on colon fluid as assessed by MRI. The study is well designed and presented and deserve publication after addressing the following minor comments

page 2 line 58: In-silico models can be also used to quantify the fluid flow and blood diffusion/reaction for drug deliver. Among these study, please cite the following:
Pasta S, Gentile G, Raffa GM, Scardulla F, Bellavia D, Luca A, Pilato M, Scardulla C. Three-dimensional parametric modeling of bicuspid aortopathy and comparison with computational flow predictions. Artif Organs. 2017 Sep;41(9):E92-E102. doi: 10.1111/aor.12866. Epub 2017 Feb 10. PMID: 28185277.

page 2 line 83: Is phase constrast MRI feasible for the colon to measseure fluid flow? please pfovide a comment on this aspect.

page 3 line 104: Please report more details on the MRI procedure (ie, acquisition time and so on)

page 8 line 252: Please consider to move Figure 5 to result section. In general, the discussion section should not present new data. Please consider that for other figure in the Discussion section. 

Author Response

We thanks the reviewers for their insight and comments on the manuscript

I have submitted a file of response to reviewers comments. There was some overlap between the reviewers so this has been submitted to both reviewers

Reviewer 2 Report

Summary and broad comments:

In this article, the authors used MRI to quantify fluid volumes within the pediatric colon. Statistical analyses were performed to evaluate the relationship between colonic fluid volume, number of fluid pockets, location of major fluid volume/pockets, and multiple statuses of the participants (most importantly the fed status and age). As the main strength, this study targeted a clinical question that has not been well studied and presented a highly feasible study method. The study of image-based quantification of pediatric colonic fluid indeed has the potential to assist in filling the knowledge gap about the physiological environment of the pediatric colon, which would further support the development of pediatric drug development. Thus, the significance of this study is clear.

However, the main limitation of this study is the small sample size and the high variability across participants. Specifically, data from 49 children are included in this study, but these data are collected from two different sites (i.e., with different scanners and acquisition protocols), and the 49 participants are different regarding multiple factors -- fed status (fluid-fed VS fasted), age range (< 2 y, 2-5 y, 6-11 y, 12-16 y), and sex. This leads to a very small number of samples within each subgroup (i.e., having the same fed status, sex, and age range). Also, the numbers of participants between different subgroups are not well balanced. All of these would potentially lead to a bias in the statistical analysis. Even though the authors took an effort to use corrected statistical tests, the bias due to the unbalanced samples is not guaranteed to be eliminated. Thus, I would consider the observations in this study as preliminary results. Further study on a larger and more coherent cohort is warranted to draw more conclusive claims.

In addition to this main concern, there are some specific comments regarding the contents (please see below). I believe this paper is suitable for publication if the comments can be addressed by the authors, and clarification be made (for example, in conclusion) to demonstrate this present study is preliminary.

Specific comments:

  1. As shown in Table 2, the group of fluid-fed participants and the group of fasted participants were completely separately collected from two sites, where different scanners and imaging protocols were used. This could lead to a systematic bias between these two groups. Please address this concern in the Discussion section.
  2. In Table 2, the row “Matrix” provides imaging matrix size for columns “UHCM” (512*512 and 256*256) but in-plane resolution for columns “BCH” (0.8*0.8mm). Please choose one description to make it coherent. The row “Echo train length” is empty for columns “BCH”. Please fill with values, or if not applicable, N/A. The row “Field of view” has “A* b cm” for columns “UHCM”. Please replace A and b with specific numbers, and make the description coherent with columns “BCH”.
  3. The text in Figure 1.D and 1.E is invisible due to low resolution. Please enlarge the panels or the text.
  4. It is hard to see anything in Figure 2.E due to low resolution. Please enlarge the panel or simply eliminate it.
  5. Results (line 179): Does the symbol “+-” referring to the standard deviation or the 95% CI? Please explicitly indicate.
  6. The axes, labels, and legends in Figure 7 are invisible due to low resolution. Please enlarge the text and make sure to provide high-resolution figures.

Author Response

We thanks the reviewer for their insight and comments

I have attached a document that answers both reviewer comments as there was some overlap
